# Body size prediction in scorpions: a phylogenetic comparative examination of linear measurements of individual body parts

Stênio Í. A. Foerster

Department of Zoology, University of Tartu, Tartu, Estonia

Corresponding author
Stênio Í. A. Foerster,
stenio.foerster@ut.ee

## ABSTRACT

Body size has always been the focus of several ecological studies due to its undeniable influence on other life-history traits. The conventional representation of body size in arthropods typically relies on linear measures, such as total body length, or the length of specific body parts that can be used to represent body size. While these measures offer simplicity over more complicated alternatives (*e.g.*, dry mass), technical problems persist for arthropods with complex body structures, as is the case for scorpions. In these animals, accurate measurements often require extensive handling, including the stretching of body parts. In light of the difficulties associated with directly measuring total length and carapace length in scorpions (two prevalent proxies for body size in the group), this study evaluates the ability of seven simple linear measurements in predicting length measures of boy size in scorpions under a phylogenetic framework. Predictive equations derived from phylogenetic mixed linear models fitted under Bayesian framework were implemented in custom R functions that can be applied for size prediction in a wide range of scorpions. Overall, accurate predictions of total length and carapace length could be achieved using any of the studied traits as single predictors. However, the most accurate predictions for total length were obtained using the length of metasomal segment V, while the best predictions for carapace length were achieved using telson length. The addition of a secondary predictor had low impact on the quality of the size predictions, indicating that increasing model complexity by incorporating additional predictors is not necessary to achieve accurate size estimates. Technical advantages and limitations associated with each linear measurement are discussed. In conclusion, this study broadens the repertoire of methods available for accurately estimating body size in scorpions, particularly in instances where body size information can only be obtained indirectly through allometric relationships.

## INTRODUCTION

Comparative analysis focusing on body size variation has gained popularity in ecological and evolutionary studies. This rise in scientific interest can be attributed to the significant role of body size in influencing and being influenced by various other life-history traits in

animals (*Peters, 1986*; *Klingenberg & Spence, 1997*; *Blanckenhorn, 2000*; *Kozłowski, Konarzewski & Czarnoleski, 2020*; *Johnson et al., 2023*). Examples of these well-known associations include relationships linking body size with range size (*Noonan et al., 2020*; *Seifert, Strutzenberger & Fiedler, 2022*), dispersal capability (*Alzate & Onstein, 2022*), mobility (*Kuussaari et al., 2014*), thermal regulation (*Gardner et al., 2011*), fecundity (*Tammaru, Esperk & Castellanos, 2002*), longevity (*Holm et al., 2022*; *Kuparinen, Yeung & Hutchings, 2023*), dietary specialization (*Seifert, Strutzenberger & Fiedler, 2023*), and sexual selection (*Janicke & Fromonteil, 2021*), among others.

Body size is typically represented by weight (mass) or length, depending on how conveniently and accurately these measures can be obtained for the taxonomic group under study. For invertebrates, particularly arthropods, measures of weight can serve as a good proxy for body size when aiming to capture both size and shape within a single value (*e.g.*, *Kendall et al., 2019*; *Ruiz-Lupión, Gómez & Moya-Laraño, 2020*; *Foerster et al., 2024a*, *2024b*; *Ude et al., 2024*), or in the analysis of metabolic rates and energy allocation (*e.g.*, *Llandres et al., 2015*; *Ferral et al., 2020*; *Turnbull, McNeil & Sinclair, 2023*). However, using weight measurements often requires handling hundreds of individuals under laboratory conditions. For instance, if dry mass is employed, animals must be sacrificed and transferred to a laboratory for proper processing, often involving specialized equipment. Moreover, body weight rather than body length is expected to be more influenced by factors such as water and fat content, nutritional condition, and reproductive stage (*Chown & Gaston, 2010*; *Stahlschmidt & Chang, 2021*). These factors can produce transient changes in body weight that may not accurately reflect the actual body size of the organism. Length measures, in contrast, provide a simpler solution to overcome such challenges. In most cases, length measures can be directly obtained in the field without the need for specialized equipment. In addition, for non-flying arthropods, length measures of body size are typically expressed as the total length of the animal without appendages, which can be thus considered a homologous trait across the studied species (*Ruiz-Lupión, Gómez & Moya-Laraño, 2020*).

Using length measures offers another advantage: specific body parts can be used to represent body size itself. This is made possible by allometry, where body parts scale in size relative to overall body size (*Pélabon et al., 2018*). By studying the allometric properties of a readily measurable trait, one can use its measurements to estimate body size or directly represent body size itself. This approach proves useful when directly measuring body size (*e.g.*, total length) is more labor-intensive, being commonplace in ecological and evolutionary studies with several invertebrate groups (*e.g.*, *Warzecha et al., 2016*; *Austin et al., 2022*; *Jahant-Miller, Miller & Parry, 2022*; *Lira et al., 2021*; *Lira, Andrade & Foerster, 2023*; *Staton et al., 2023*). Not surprisingly, significant efforts have been directed towards developing predictive models (equations) of body size based on meristic traits that are simpler to measure (*Kendall et al., 2019*; *Foerster et al., 2024a*). This includes "classical" linear predictive equations on a log-log scale (*Huxley, 1932*; *Packard, 2013*; *Pélabon et al., 2018*), which are commonly employed to predict body size in

various insect groups, including (but not limited to) Coleoptera, Isoptera, Thysanura (*Ruiz-Lupión, Gómez & Moya-Laraño, 2020*), Hymenoptera (*Kendall et al., 2019*; *Ruiz-Lupión, Gómez & Moya-Laraño, 2020*), Lepidoptera (*Foerster et al., 2024a*), Diptera (*Kendall et al., 2019*) and, to a lesser extent, in other arthropods, such as arachnids (*Hódar, 1996*, *1997*; *Höfer & Ott, 2009*; *Shiao et al., 2019*).

Simpler proxies and predictive models for estimating body size are especially valuable for arachnids, especially those belonging to clades with complex body architecture. Directly measuring the total length of scorpions, for instance, can be methodologically challenging due to their overall body structure. The nearly 2,830 extant scorpion species described to date (*Rein, 2024*) share a characteristic body structure consisting of three main segments: the prosoma, mesosoma, and metasoma. Obtaining precise measurements of total length in scorpions requires stretching anatomical parts such as the metasoma, commonly referred to as "tail", which demands considerable time and expertise. Procedures like these are impractical to be performed on live specimens, being also problematic for museum specimens without the considerable risk of causing damage to the individual, which in turn, affects the precision of the measurements. This is likely one of the reasons why carapace length is frequently used as a proxy for body size in scorpions (*e.g.*, *DeSouza et al., 2016*; *Seiter & Stockmann, 2017*; *Lira, Andrade & Foerster, 2023*; *Moreira et al., 2022*; *Giménez Carbonari et al., 2024*), despite the lack of empirical evidence demonstrating its suitability as an indicator of body size, at least in comparative contexts. However, even carapace length cannot always be easily measured. In some instances, the carapace itself can be damaged, or its edges may be covered by other body parts, such as the pedipalps, necessitating handling that also increases the risk of structural damage to the specimen. Therefore, models that enable accurate prediction of body size measurements in scorpions (*e.g.*, total length, carapace length) from simpler linear measurements of various body parts are valuable tools, particularly when dealing with highly damaged or fragile specimens. In some cases, these models represent the only alternative to obtaining information on body size, such as with old and poorly preserved specimens from scientific collections and some fossil records.

The aim of this study is to establish coefficients for accurately predicting body size in scorpions, while considering their evolutionary history. I present a series of simple linear equations implemented in custom R functions that can be used to effectively predict total length and carapace length—two commonly used measures of body size in scorpions (*e.g.*, *Outeda-Jorge, Mello & Pinto-da-Rocha, 2009*; *Seiter & Stockmann, 2017*; *Lira et al., 2018*; *Lira, Andrade & Foerster, 2023*; *Giménez Carbonari et al., 2024*) based on simple linear measurements of individual body parts. Specifically, I assessed the predictive power of seven linear measurements using simple and multiple phylogenetic mixed linear models across 195 scorpion species from the family Buthidae. The family Buthidae is the most species-rich clade within the order Scorpiones, offering sufficient morphometric variation to effectively represent the order in terms of overall body size as well as the size and shape of individual body parts.

## MATERIALS AND METHODS

### Trait data and species

Linear measurements were primarily obtained from a comprehensive list of taxonomic and ecological studies on buthid scorpions (Supplemental file S1). The measurements were mainly presented in simple tables that could be manually retrieved from the selected articles. The present study acknowledges that while measurement errors in the selected traits can be minimized, they cannot be entirely eliminated. However, it is assumed that these errors do not significantly impact the quality of size predictions, as (1) the articles selected for extracting the meristic information were mostly authored by experienced scorpion taxonomists, and (2) the models are intended to provide an average estimate of body size rather than precise, exact predictions.

Two proxies for body size were employed in this study (analyzed separately, see below): total length (toL), measured from the anterior margin of the carapace to the aculeus tip (Fig. 1), and carapace length (carL), expressed by the maximum distance between the anterior and the posterior margins of the carapace (Fig. 1). The candidate traits used as predictors of the two body size measures are illustrated in Fig. 1. These included: chela length (cheL) measured from the base of the manus to the tip of the fixed finger, chela width (cheW) measured dorsally at the middle of the manus, telson length (teL) obtained from the distance connecting the base of the vesicle to the aculeus tip, telson width (teW) corresponding to the width of the vesicle in dorsal view, and the length (met5L) and width (met5W) of the metasomal segment V (Fig. 1; *Sissom, Polis & Watt, 1990*); the ability of carL in predicting toL was also evaluated. The (predictor) measurements were chosen because they are relatively easy to measure in well-preserved specimens and readily available in taxonomic studies of scorpions (Supplemental file S1).

Raw trait values were used to calculate species means separately for males and females for species represented by more than one individual per sex. On average, two individuals per sex were used to calculate species means for each species (Supplemental file S1). The proportion of missing data in the trait matrices ranged from 2% (cheL in both sexes) to 13% (teW in males). Missing predictor trait values in both matrices were imputed using the missForest R package (*Stekhoven & Bühlmann, 2012*) due to its accuracy and computational efficiency (*Penone et al., 2014*; *May, Feng & Adamowicz, 2023*). Imputation was performed separately for each sex using the log-transformed species means. Phylogeny was incorporated by including the first 10 eigenvectors from a principal coordinate analysis on cophenetic distances (*Penone et al., 2014*) calculated from an ultrametric phylogeny generated in this study (see below) in each trait matrix.

A second analysis was performed to assess the precision of the imputation procedure on each trait matrix (one per sex). It consisted in pruning the trait matrices to retain only species with no missing data ($n$ = 119) in the original scale (mm). Then, 13% of data—the maximum proportion of missing data observed in the original matrices—was randomly assigned as missing data in each trait in the pruned matrices. The imputation was performed on the pruned matrices with their respective phylogenetic eigenvectors, and the imputation accuracy was assessed using the normalized root mean square error (NRMSE),

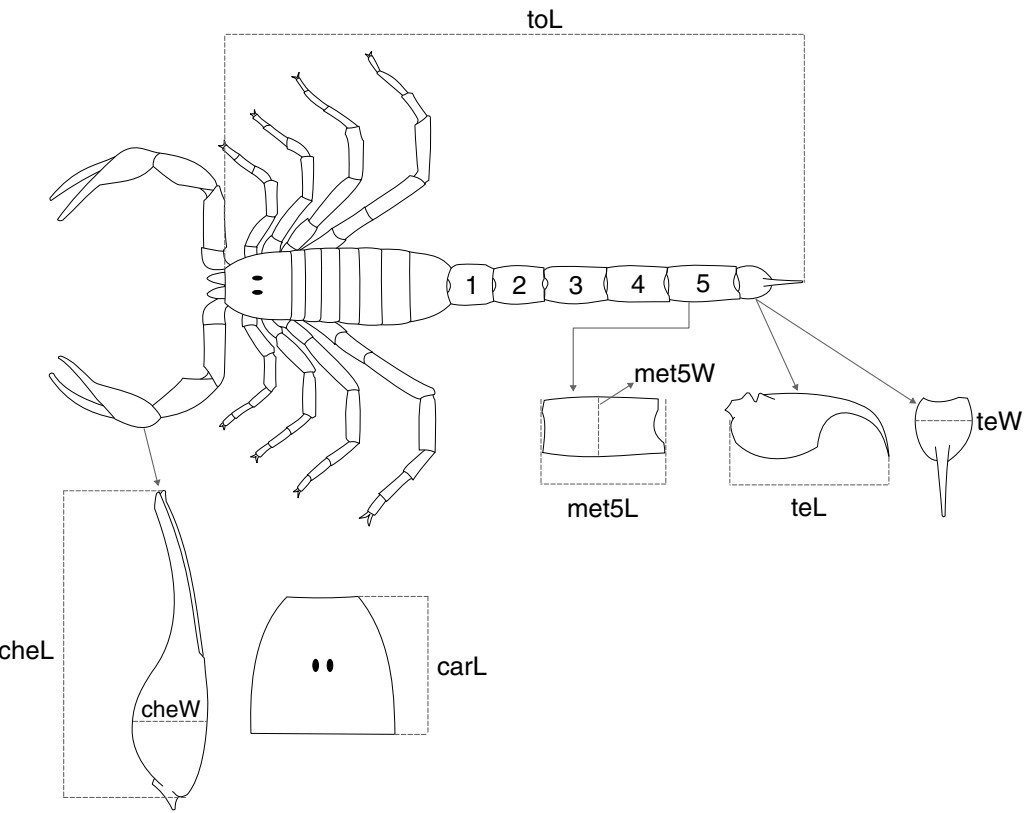

**Figure 1 Schematic representation showing the overall morphology of a scorpion and the meristic traits collected for buthid species.** The two variables used to express body size were (toL) total length, and (carL) carapace length. The following traits were used to predict the two proxies for body size: (cheL) chela length, (cheW) chela width, (met5L) length of the metasomal segment V, (met5W) width of the metasomal segment 5, (teL) telson length, and (teW) telson width.

which compares predicted and actual values, with values closer to zero indicating higher accuracy (*Stekhoven & Bühlmann, 2012*). The imputation on the pruned matrices was repeated 10 times for each sex, and the mean NRMSE (in mm) was reported for each.

## Phylogeny

Shared evolutionary history between species were considered by using an ultrametric phylogenetic tree for the studied species. The tree was derived from nucleotide sequences of four DNA loci commonly used in systematic studies of scorpions (*e.g.*, *Esposito et al., 2018*; *Esposito & Prendini, 2019*; *Štundlová et al., 2022*): partial sequences of the mitochondrial genes cytochrome c oxidase I (COI), 16S rRNA, and partial sequences of the nuclear genes 18S rRNA and 28S rRNA. DNA data were primarily sourced from GenBank (Supplemental file S1), with supplementary sequences obtained at the LABBE/UFPE (Recife, Brazil) as indicated in Supplemental file S1. DNA extraction, purification, amplification, and sequencing procedures to generate supplementary sequences were conducted following the methods described by *Esposito et al. (2018)*. Chromatograms were

processed with Pregap4 (*Staden, Beal & Bonfield, 1999*) for the assembly of consensus sequences, which were subsequently deposited in GenBank (Supplemental file S1).

Sequences of the four loci were aligned independently using MAFFT v.7 (*Katoh & Standley, 2013*) with default settings; badly aligned regions were automatically removed using Gblocks v 0.91b (*Castresana, 2000*) with default settings. Gene sequences were then concatenated using the Concatenator software (*Vences et al., 2022*), resulting in a final alignment block of 2,836 base pairs for 268 species, with the selection of the bothriurid species *Brachistosternus paposo* Ojanguren Affilastro & Pizarro-Araya, 2014 as the sole outgroup.

Tree topology and branch lengths were estimated by maximum likelihood in IQ-TREE v.2.3.2 (*Minh et al., 2020*), with branch support calculated using 1,000 ultrafast bootstraps (*Hoang et al., 2018*), and the SH-aLRT test (-alrt 1,000; *Guindon et al., 2010*). For the COI sequences, partitioning was based on codon position, while the remaining (ribosomal) genes were partitioned by gene. The optimal partitioning scheme for the genes was determined using ModelFinder (*Chernomor, Von Haeseler & Minh, 2016*; *Kalyaanamoorthy et al., 2017*) in IQ-TREE (-m MF+MERGE), which also identified the best-fit evolutionary model for each partition (Information S1) based on evolutionary substitution models available in BEAST v1.10.4 (*Suchard et al., 2018*). The resulting maximum likelihood tree was converted to ultrametric (branch lengths in million years) using the penalized likelihood method implemented in the chronos function from the ape R package (*Paradis & Schliep, 2019*). To do so, nine calibration points were established based on fossil records and previous node age estimations (Information S1). The "clock" model was used in the chronos function as it had the lowest score for penalized hierarchical information criterion among the available models (*Paradis & Schliep, 2019*). Subsequently, node ages were estimated for the ultrametric phylogeny using Bayesian inference with fixed tree topology operators in BEAST. Lognormal priors were defined to align with the calibration points used in IQ-TREE. The parameters meanlog and sdlog of the lognormal distributions were optimized, ensuring that the 2.5% and 97.5% quantiles of each distribution matched the age intervals of the respective calibration point. The custom R function for defining these optimal lognormal priors is available in Supplemental file S1, along with a brief explanation and tutorial. The Markov chain Monte Carlo (MCMC) sampling in BEAST was configured with two chains, each consisting of 50 million iterations, sampled every 1,000 iterations, with a 25% burn-in. The resulting log and tree files were combined using LogCombiner 1.10.4 (*Suchard et al., 2018*) and analyzed in Tracer 1.7.2 (*Rambaut et al., 2018*). No convergence issues were detected for all estimated parameters (ESS > 200).

## Data analysis

All analyses were conducted using R software (*R Core Team, 2024*), with natural logarithmic transformation applied to all linear measurements. The phylogenetic tree was pruned using the function drop.tip in the ape R package to retain only the species presented in the trait data ($n = 195$). To better understand the covariation structure of the trait data, considering evolutionary dependence, a phylogenetic principal component

analysis (PCA) was carried out using the function phyl.pca and the Pagel's λ method (*Pagel, 1997*, *1999*) as implemented in the phytools R package (*Revell, 2024*). The phylogenetic PCA facilitated the evaluation of how effectively the candidate predictors captured size information within the trait space (*e.g.*, *Konuma & Chiba, 2007*; *Konuma, Nagata & Sota, 2011*; *Foerster et al., 2024a*).

The following analysis focused on assessing the predictive ability of each linear measurement for estimating both toL and carL. This was achieved using phylogenetic mixed linear models implemented in the MCMCglmm v.2.36 R package (*Hadfield, 2010*). By including species-level (repeated measures) random intercept factors, alongside the inverse of phylogenetic covariance matrix, the mixed models enabled testing the interaction effects between each predictor trait and sex, taking phylogeny into account. The model coefficients were estimated using Bayesian inference with default priors in MCMCglmm; a single MCMC chain with 50,000 iterations sampled every 50 was set for each model, with a burn-in of 5,000 iterations. Such parameters were enough to ensure proper convergence and mixture of MCMC chains, as evidenced by high effective sample sizes (see Results). The performance of the predictive models was assessed based on the root-mean-square error (RMSE), which represents the average difference (in mm) between predicted and actual values, with the best model presenting the lowest associated RMSE value. Only one predictor was tested at a time, allowing the identification of the most effective single predictor of toL and carL—*i.e.*, the model with the lowest RMSE. After finding the best single predictor for each body size measurement, a second predictor was added to the best-fit simple regression model to explore whether increasing model complexity resulted in substantial improvement in prediction accuracy. Only one secondary predictor was added at time, ensuring that the resulting multiple regression model contained only two predictors. Subsequently, the accuracy of body size predictions was compared between the simple and the multiple regression models using their respective RMSE values. Based on the results obtained with buthids, where the best predictions of body size were achieved with simple regression models (see results), the best-fit simple regression model was applied to predict toL and carL in 30 randomly selected non-buthid scorpions. Deviances, calculated as the difference between predicted and actual values of toL and carL, were reported for non-buthids.

## RESULTS

### General findings

Total length ranged from 14.9 to 116.2 mm in females (mean ± standard deviation: 53.7 ± 20.8 mm) and 11.9 to 119.1 mm in males (50.6 ± 20.6 mm). Measurements of carapace length ranged from 2 to 12.5 mm in females (5.9 ± 2.2 mm) and 1.6 to 10.9 mm in males (5.3 ± 2.1 mm). The phylogenetic trait imputation performed on the missing values for predictor traits proved to be highly accurate, with mean NRMSE (calculated across 10 iterations) of 0.01 mm for both sexes. The visual inspection of the trait space through phylogenetic PCA indicated high correlation between the studied traits (Figs. 2A, 2B), which was corroborated by the high trait loadings on the phylogenetic PC1 (Table 1). Specifically, all traits correlated more strongly (and in the same direction) with PC1,

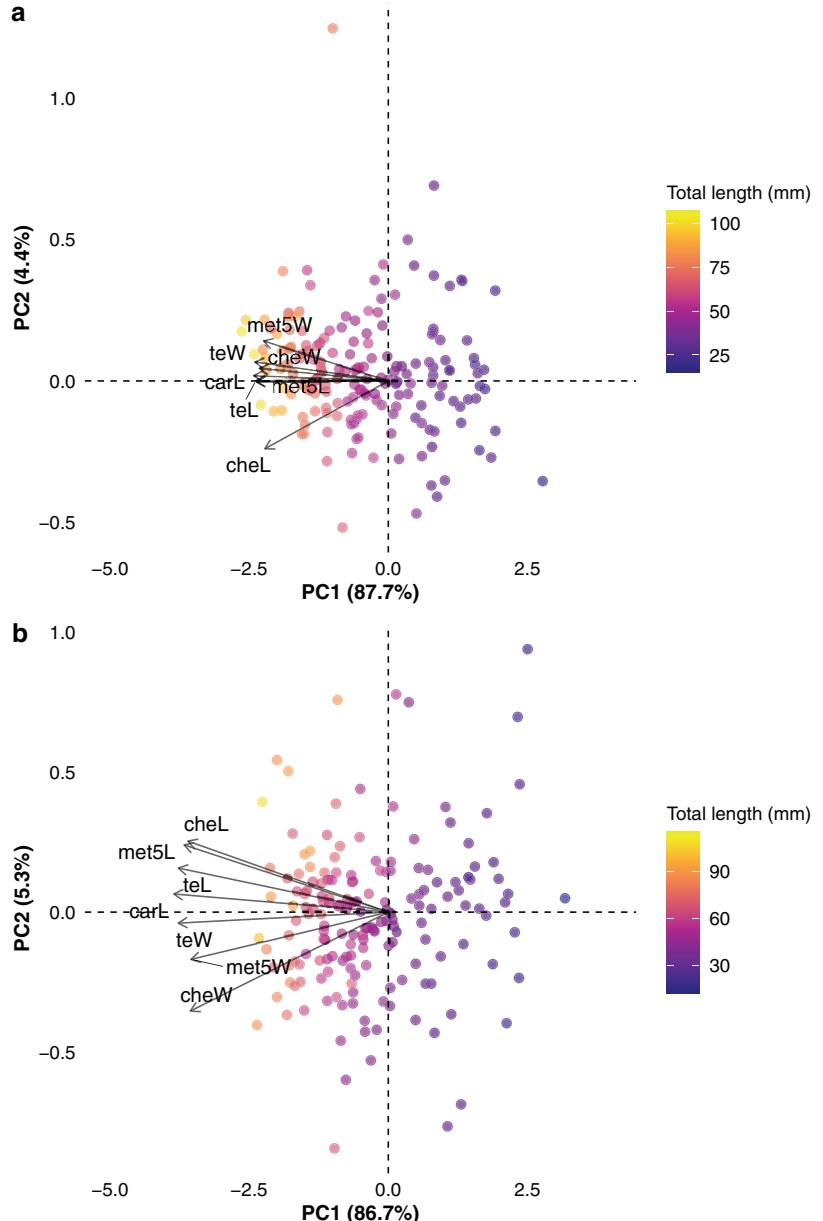

**Figure 2 Phylogenetic principal component analysis conducted with (A) female and (B) male buthid scorpions, illustrating the arrangement of species (points) and examined traits (arrows) across the two principal components.** Refer to Fig. 1 for the abbreviations of traits. The gradient in size along axis 1 is depicted by the proximity of trait vectors to axis 1 and the linear variation in total length values across axis 1. The analysis was performed using natural logarithms of trait values, taking phylogenetic relationships among species into account.

indicating that PC1 effectively captured size (length) information. The strength and direction of trait correlations in relation to PC1 indicated a size gradient along this axis. Smaller species were primarily located on the right side of the size gradient (positively correlated with PC1), while larger species occupied the left side of the gradient (negatively correlated with PC1). Species appeared more dispersed in the two-dimensional trait space

**Table 1 Trait loadings on the first two axes of the phylogenetic principal component analysis performed on linear measurements of body parts of buthid scorpions ($n = 195$).**

| Trait | Female | | Male | |
|---|---|---|---|---|
| | PC1 | PC2 | PC1 | PC2 |
| Carapace length | −0.97 | 0.03 | −0.98 | 0.07 |
| Chela length | −0.89 | −0.43 | −0.91 | 0.26 |
| Chela width | −0.93 | 0.08 | −0.90 | −0.36 |
| Length of metasomal segment V | −0.95 | −0.01 | −0.93 | 0.25 |
| Width of metasomal segment V | −0.90 | 0.25 | −0.90 | −0.17 |
| Telson length | −0.96 | 0.0004 | −0.96 | 0.16 |
| Telson width | −0.96 | 0.12 | −0.96 | −0.04 |
| Variance (%) | 87.70 | 4.40 | 86.70 | 5.30 |

**Note:**
Phylogenetic dependence in trait values was accounted for by using the lambda method. All trait values were log-transformed before the analysis.

of male data (Fig. 2B) compared to the trait space constructed from female data. Additionally, smaller species tended to be more widely distributed in the bidimensional trait space for both sexes, though this pattern was more pronounced in males (Fig. 2B). The tree topology and node age estimations were largely consistent with previous phylogenetic hypotheses (*e.g.*, *Ojanguren-Affilastro et al., 2017*; *Santibáñez-López et al., 2022*; *Štundlová et al., 2022*), recovering major buthid clades, such as the *Tityus* and *Buthus* groups, with strong statistical support (Supplemental file S1). The monophyly of species-rich genera such as *Tityus* and *Centruroides* was also well supported (Supplemental file S1). Divergence times were estimated at 57.7 Mya (95% HPD: 53–63 Mya) for the *Tityus* group and 43.2 Mya (39.7–46.6 Mya) for the *Buthus* group.

## Predicting total length

Overall, all studied traits predicted toL with satisfactory accuracy, with RMSE values not exceeding 12.2 mm (Table 2). The most effective single predictor of toL was met5L (RMSE = 6.4), closely followed by carL (Table 2). The linear relationship between met5L and toL was consistent across males and females (Fig. 3A), further supported by the non-significant interaction between met5L and sex, as observed for all other traits used to predict toL (Supplemental file S1). Conversely, met5W proved to be the least accurate single predictor (Table 2), aligning with its lower loadings on the first axis of the phylogenetic PCA, evidencing its limited ability to represent overall size (length) compared to the other traits (Table 1). The coefficients obtained from the model including met5L proved to be efficient in predicting toL in non-buthid scorpions, with differences between predicted and actual values not exceeding 7.5 mm (Table 3).

The multiple regression analyses revealed that total length predictions could be slightly improved by adding a secondary predictor alongside met5L (Supplemental file S1). However, these improvements were modest: the difference in RMSE between the best-fit multiple regression model (met5L + cheW, RMSE = 5.13 mm) and the best single-predictor model was only 1.28 mm.

**Table 2 The performance of linear measurements in predicting (ToL) total length and (CarL) carapace length in buthid scorpions (*n* = 195) was assessed using Bayesian phylogenetic mixed linear models.**

| Response | Predictor | RMSE | Coefficients | Mean | Lower | Upper | ESS |
|---|---|---|---|---|---|---|---|
| toL | met5L | 6.41 | Intercept | 2.13 | 2.03 | 2.22 | 900 |
| | | | Slope | 0.95 | 0.91 | 0.99 | 795 |
| | carL | 6.46 | Intercept | 2.18 | 2.08 | 2.27 | 900 |
| | | | Slope | 1.01 | 0.97 | 1.05 | 900 |
| | teL | 7.22 | Intercept | 2.32 | 2.21 | 2.44 | 1,088 |
| | | | Slope | 0.90 | 0.86 | 0.94 | 900 |
| | cheL | 8.84 | Intercept | 2.27 | 2.08 | 2.50 | 900 |
| | | | Slope | 0.72 | 0.67 | 0.78 | 900 |
| | teW | 9.27 | Intercept | 3.29 | 3.13 | 3.44 | 994 |
| | | | Slope | 0.82 | 0.77 | 0.86 | 900 |
| | cheW | 10.70 | Intercept | 3.40 | 3.23 | 3.57 | 900 |
| | | | Slope | 0.63 | 0.57 | 0.69 | 900 |
| | Met5W | 12.24 | Intercept | 3.13 | 2.94 | 3.34 | 678 |
| | | | Slope | 0.71 | 0.65 | 0.77 | 805 |
| carL | teL | 0.68 | Intercept | 0.24 | 0.13 | 0.34 | 800 |
| | | | Slope | 0.84 | 0.80 | 0.87 | 752 |
| | teW | 0.76 | Intercept | 1.12 | 0.99 | 1.22 | 810 |
| | | | Slope | 0.78 | 0.74 | 0.81 | 1,090 |
| | Met5L | 0.96 | Intercept | 0.10 | −0.02 | 0.22 | 900 |
| | | | Slope | 0.85 | 0.80 | 0.90 | 900 |
| | cheL | 0.98 | Intercept | 0.22 | 0.04 | 0.42 | 900 |
| | | | Slope | 0.66 | 0.61 | 0.71 | 900 |
| | Met5W | 1.02 | Intercept | 0.95 | 0.75 | 1.09 | 870 |
| | | | Slope | 0.71 | 0.66 | 0.76 | 900 |
| | cheW | 1.06 | Intercept | 1.22 | 1.08 | 1.39 | 1,156 |
| | | | Slope | 0.60 | 0.55 | 0.65 | 900 |

**Note:**
See Fig. 1 for trait abbreviations.
All variables were log-transformed (using natural logarithms) before the analysis. Repeated measurements for species (*i.e.*, one measurement per sex) were included as random intercept factors in the models. The equation for predicting the desired body size metric (in mm) is given by: $Y = \exp(a + \ln(x)b)$, where a is the intercept, x is the predictor, and b is the slope of the predictor. RMSE represents the residual mean squared error, lower and upper 95% credible intervals are presented. ESS, effective sample size.

## Predicting carapace length

The simple phylogenetic regression models identified teL as the best single predictor of carL (Table 2), with a common slope coefficient across sexes (Fig. 3B, Supplemental file S1). cheW, on the other hand, was the least accurate predictor of carL, yielding an RMSE of 1.1 mm (Table 2). The interactions between predictor traits and sex were largely non-significant when predicting carL, with the exception of the sex*met5L interaction (Supplemental file S1). However, the model incorporating this interaction did not outperform the best simple regression model for carL: the RMSE for the interaction model was 0.9 mm, compared to 0.7 mm when using teL alone. Predictions of carL based on teL
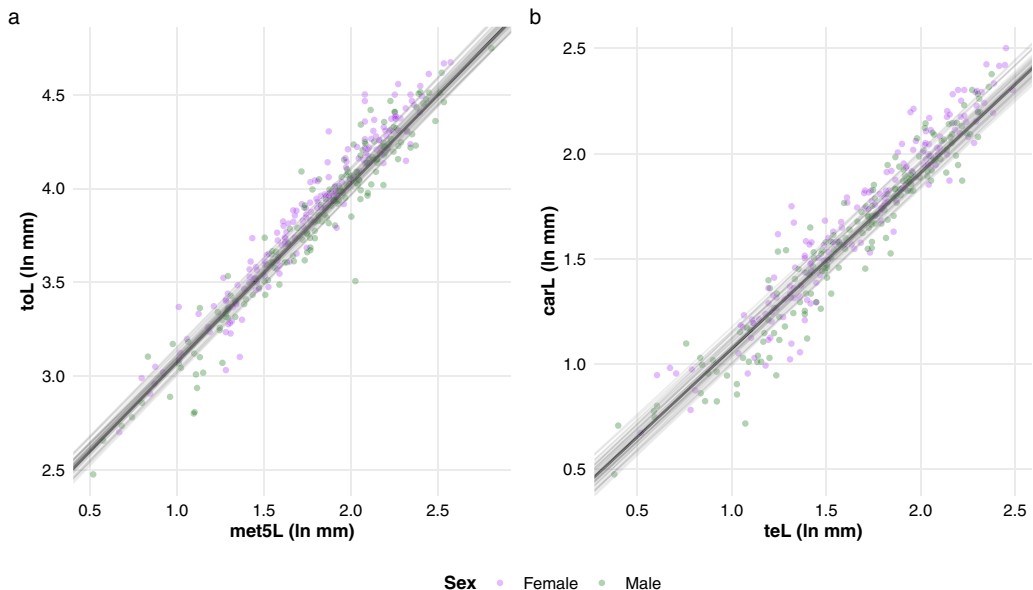

**Figure 3** Linear relationships between (A) total length and the length of metasomal segment V, and (B) carapace length and telson length in buthid scorpions (*n* = 195) were estimated using Bayesian phylogenetic mixed linear models. The thick line represents the slopes of these relationships, calculated from the mean posterior estimates (50,000 MCMC samples). The thin lines indicate the 95% credible intervals of the slope coeûcients, derived from 50 random MCMC samples from each model. All variables were log-transformed (natural logarithms) before the analysis.

**Table 3 Size predictions for non-buthid scorpions (*n* = 30) according to the best-fit phylogenetic simple regression models fitted under a Bayesian framework.**

| Species | Family | Sex | ToL | ToL Dev. | CarL | CarL Dev. | Source |
|---|---|---|---|---|---|---|---|
| *Vaejovis vorhiesi* | Vaejovidae | Female | 28.25 | −0.36 | 3.45 | −0.09 | *Ayrey & Soleglad (2015)* |
| *Orobothriurus grismadoi* | Bothriuridae | Male | 30.36 | −0.61 | 3.47 | 0.85 | *Ojanguren Affilastro et al. (2009)* |
| *Scorpiops kovariki* | Scorpiopidae | Female | 32.95 | −0.68 | 4.84 | −0.39 | *Tang et al. (2024)* |
| *Chaerilus seiteri* | Chaerilidae | Male | 24.20 | −0.80 | 3.60 | 0.19 | *Kovařík (2012b)* |
| *Teuthraustes braziliensis* | Chactidae | Male | 52.20 | −1.24 | 7.50 | −0.49 | *Lourenço & Duhem (2010)* |
| *Calchas kosswigi* | Iuridae | Female | 34.25 | −1.53 | 4.30 | 0.33 | *Yağmur et al. (2013)* |
| *Typhlochactas mitchelli* | Typhlochactidae | Male | 8.99 | 1.68 | 1.17 | 0.66 | *Sissom (1988)* |
| *Diplocentrus izabal* | Diplocentridae | Male | 53.85 | 1.79 | 6.75 | −0.71 | *Armas & Trujillo (2016)* |
| *Scorpiops tongtongi* | Scorpiopidae | Female | 41.90 | 1.82 | 6.70 | −0.85 | *Tang (2022a)* |
| *Megacormus franckei* | Euscorpiidae | Female | 42.48 | −2.12 | 6.38 | −0.53 | *Kovařík (2019)* |
| *Euscorpius gulhanim* | Euscorpiidae | Male | 26.96 | −2.13 | 3.97 | −0.03 | *Yağmur (2024)* |
| *Diplocentrus franckei* | Diplocentridae | Male | 57.80 | 2.50 | 7.10 | −2.06 | *Santibáñez-López (2014)* |
| *Balsateres cisnerosi* | Vaejovidae | Male | 47.00 | 2.52 | 6.10 | −0.10 | *González-Santillán & Prendini (2018)* |
| *Vaejovis troupi* | Vaejovidae | Female | 25.70 | 2.94 | 3.40 | 0.04 | *Ayrey & Soleglad (2015)* |
| *Hadrurus anzaborrego* | Hadruridae | Male | 84.90 | −3.32 | 11.45 | −1.38 | *Soleglad, Fet & Lowe (2011)* |
| *Calchas anlasi* | Iuridae | Male | 28.55 | 3.43 | 3.70 | 0.60 | *Yağmur et al. (2013)* |

(Continued)

| Species | Family | Sex | ToL | ToL Dev. | CarL | CarL Dev. | Source |
|---|---|---|---|---|---|---|---|
| *Chactas moreti* | Chactidae | Male | 43.10 | 3.52 | 5.60 | 0.40 | *Lourenço (2014)* |
| *Scorpiops deshpandei* | Scorpiopidae | Male | 54.24 | −3.71 | 8.43 | −1.46 | *Tang et al. (2024)* |
| *Urophonius trewanke* | Bothriuridae | Male | 32.53 | 3.73 | 3.71 | 1.44 | *Ojanguren-Affilastro et al. (2024)* |
| *Euscorpiops thaomischi* | Scorpiopidae | Male | 45.00 | 3.79 | 7.80 | −0.86 | *Kovařík (2012a)* |
| *Alpiscorpius lambda* | Euscorpiidae | Male | 21.13 | 3.81 | 3.52 | 0.09 | *Kovařík et al. (2019)* |
| *Spinochactas mitaraka* | Chactidae | Female | 12.92 | 4.33 | 2.00 | 0.54 | *Lourenço (2016)* |
| *Qianxie solegladi* | Pseudochactidae | Female | 24.70 | 5.05 | 3.10 | 0.69 | *Tang (2022b)* |
| *Brachistosternus mattonii* | Bothriuridae | Male | 54.46 | −5.23 | 5.74 | 0.66 | *Ojanguren-Affilastro (2005)* |
| *Hadruroides tishqu* | Caraboctonidae | Female | 57.90 | 5.26 | 7.60 | 0.39 | *Ochoa & Prendini (2010)* |
| *Bothriurus delmari* | Bothriuridae | Female | 25.90 | 5.34 | 3.70 | 0.43 | *Santos-Da-Silva, Carvalho & Brescovit (2017)* |
| *Hadrurus anzaborrego* | Hadruridae | Female | 104.30 | −5.57 | 13.20 | −1.66 | *Soleglad, Fet & Lowe (2011)* |
| *Chaerilus solegladi* | Chaerilidae | Male | 45.00 | 5.96 | 6.40 | 0.23 | *Kovařík (2012b)* |
| *Diplocentrus izabal* | Diplocentridae | Female | 52.25 | −6.72 | 7.50 | −1.69 | *Armas & Trujillo (2016)* |
| *Troglotayosicus akaido* | Troglotayosicidae | Male | 19.53 | 7.47 | 3.32 | 0.33 | *Moreno-González, Luna-Sarmiento & Prendini (2024)* |

**Note:**
Deviances were calculated as the difference between predicted and actual values of toL and carL. All measurements are reported in millimeters.

were also fairly accurate for non-buthid scorpions, with the differences between predicted and actual values generally remaining under 1 mm (Table 3).

Again, adding a secondary predictor alongside teL provided no substantial improvement in model accuracy. The difference in RMSE between the best simple regression model and the best multiple regression model (teL + met5W) was only 0.1 mm when predicting carL (Table 2, Information S1).

## Implementation

All predictive equations presented in this study are implemented in four R functions, allowing for the prediction of toL and carL using either single predictors or a combination of two predictors. These functions are provided in Supplemental file S2 (scorpion_sizeR. R). After downloading the file to the working directory in R, the functions can be loaded into the R environment by using the command: source ("scorpion_sizeR.R").

The functions simple_toL and simple_carL are used for predicting toL and carL based on single predictors. The user provides measurements (in mm) using the values_mm argument and specifies the predictor trait *via* the predictor argument, which can take one of the following values: "met5l", "met5w", "carl" (for simple_toL only), "tel", "tew", "chel", and "chew".

The functions multi_toL and multi_carL predict toL and carL using multiple regression models. For multi_toL, the user needs to supply the length of the fifth metasomal segment (met5l_mm), along with the value and name of the secondary predictor *via* the arguments secPred_mm and second_predictor, respectively. For multi_carL, the user provides teL (tel_mm) and the secondary predictor in the same way as for multi_toL. All measurements should be in millimeters; predicted values are also returned in millimeters.

## DISCUSSION

This study explored the ability of linear measurements of specific body parts in predicting the overall body size in buthid scorpions taking phylogenetic relationships among species into account. As a result, several predictive equations that accurately and easily estimate body size metrics such as toL and carL, which are commonly used to express body size in scorpions (*e.g.*, *Fox, Cooper & Hayes, 2015*; *McLean, Garwood & Brassey, 2018*), are reported and implemented in custom R functions.

The general lack of statistically significant interactions involving sex indicated that the predictive models can be generalized for size prediction in both male and female scorpions. This is particularly useful in cases where species exhibit subtle morphological differences between the sexes or when specimens are heavily damaged or fragmented in a way that the determination of sex is impossible. The latter scenario may be common in various fields, including paleontology where only partial remains may be available or visible for performing precise measurements (*e.g.*, *Santiago-Blay et al., 2004*; *Riquelme et al., 2015*; *Lourenço, 2023*), and studies of feeding ecology, where only parts of the specimens may be recovered (*e.g.*, *Sahley et al., 2015*; *Nordberg et al., 2018*; *Karawita et al., 2020*; *Qashqaei, Ghaedi & Coogan, 2023*). Moreover, the predictive equations presented here can also be valuable for obtaining precise measurements of body size for scorpions stored in scientific collections, as the conventional measurement can be challenging without risking damage to the specimen.

The simple phylogenetic regression models indicated that carL was an effective predictor and good proxy for body size (toL), albeit not the best. Although this finding may not sound surprising given that several studies have used carL as an indicator of body size in scorpions (*e.g.*, *DeSouza et al., 2016*; *Seiter & Stockmann, 2017*; *Moreira et al., 2022*; *Lira, Andrade & Foerster, 2023*; *Giménez Carbonari et al., 2024*), it constitutes the first empirical validation of the suitability of carL in predicting total length and also as being a reliable proxy for body size in these animals, tested with robust comparative data. With a few exceptions (*McLean, Garwood & Brassey, 2018*) the use of carL as an indicator of body size is not always clearly justified. From a technical perspective, using carL offers some useful methodological advantages over toL, the most straightforward of them being the ease of measure. Ideally, however, such methodological advantages should be supported by empirical validations obtained from compelling evidence, which the present study now provides.

The visual inspection of trait space provided by the phylogenetic principal component analysis indicated no major differences in the covariation structure of the trait data between sexes, aligning with the general absence of sex-specific slopes in the linear relationships between each trait and the two body size metrics. However, the phylogenetic PCA revealed that species were more widely dispersed in the two-dimensional trait space constructed from male data, particularly for smaller species. This suggests that morphological diversity, in a multivariate context, may be greater in males, supporting previous hypotheses that link sexual selection as a key driver of sexual size dimorphism in scorpions, particularly influencing the size and shape of male body parts (*Lira et al., 2018*;

*McLean, Garwood & Brassey, 2018*; *Visser & Geerts, 2021*, but see *Sánchez-Quirós, Arévalo & Barrantes, 2012*). Besides sexual selection, other selective forces such as predation pressure and resource exploitation are also correlated with body size, in which smaller species are generally more vulnerable to negative impacts, including higher predation rates (*Moreira et al., 2022*) and foraging constraints (*Polis & McCormick, 1987*). These adverse effects are likely to be more pronounced in male scorpions, as they tend to be more active on the surface while foraging and seeking mates (*Polis, 1990*). Therefore, it is plausible that the greater overall dispersion observed in male trait space, especially among the smallest species, reflects, to some extent, these selective forces influencing species morphology.

The size gradient observed along the phylogenetic PC1, supported by the observed trait loadings, suggests that not only do length-related traits contain significant information about overall size, but also traits related to the widths of the chela and telson. These findings indicate that while obtaining a reasonable indicator of size (length) in scorpions is relatively easy, the same cannot be said for reliably capturing information on overall shape. In such a context, morphometric ratios between different body parts or size-normalized trait measurements might offer alternative means of capturing shape-related information. Some morphometric ratios have been utilized as diagnostic characters in a taxonomic context (*e.g.*, *Soleglad & Fet, 2010*; *Kovařík & Lowe, 2022*, but see *González-Santillán & Prendini, 2015*) or in intraspecific analyses of morphological variation (*Alqahtani et al., 2022*). However, for the purposes of this study, the choice was made to prioritize the primary goal of proposing simple predictive equations capable of accurately estimating body size without the necessity of measuring multiple body parts or calculating secondary variables (*i.e.*, morphometric ratios) for size prediction. Still, the arrangement of trait values within the two-dimensional space of the phylogenetic PCA, together with the magnitude and direction of trait loadings on axis 1, suggests that allometry likely plays a crucial role in determining the size of the traits examined in this study. However, the phylogenetic PCA is unable to clearly reveal the specific allometric patterns associated with each trait—for instance, whether they follow isometry, positive, or negative allometric patterns. Therefore, the findings reported here serve as a foundation for further investigations in this area, which could offer valuable insights not only into the evolutionary forces (such as sexual and natural selection) shaping the morphology of scorpions but also into the evolution of sexual size dimorphism and sexual body component dimorphism (*Fox, Cooper & Hayes, 2015*) in these animals.

Regarding the predictive ability of the studied traits, the simple phylogenetic regression models indicated that met5L was the best single predictor of toL. Met5L is particularly advantageous for measurement because assessing it typically does not require fully extending the metasoma or manipulating other body parts for obtaining the measurements. Furthermore, metasomal segments are highly sclerotized structures in scorpions, which makes them less prone to breakage compared to other anatomical parts such as the aculeus (part of the telson)—it is not uncommon to encounter individuals with broken aculeus tip (*e.g.*, *Armas, 1999*; *Teruel & Rein, 2010*; *Teruel & Kovařík, 2014*). This issue might be more concerning for the utilization of teL as a predictor of body size, particularly in smaller species because the accuracy of predicting carapace or total length

may be compromised in specimens with broken aculeus. The degree of this artifact will vary depending on the amount of aculeus missing relative to the approximate length of the telson. For example, if one considers a hypothetical scenario where an aculeus breakage results in the loss of 1 mm length in the telson of an adult specimen of *Centruroides exilimanus Teruel & Stockwell, 2002*, it will represent, on average, only 9% of the telson length in this species (average telson length = 10.8 mm, *Teruel & Stockwell, 2002*; *Viquez & Armas, 2005*). In contrast, the same damage would correspond (on average) to 68% of the telson length in a specimen of *Microtityus fundorai Armas, 1974* (telson length = 1.46 mm, *Armas, 1974*). Fortunately, all other predictors performed relatively well in estimating both toL and carL. Hence, the general recommendation is to use teL as a predictor of body size only when the telson is fully intact. This is particularly relevant for predicting carL, where teL proved to be the most accurate predictor. In such cases, it might be more advantageous to use teW, as the vesicle (the anatomical region from which telson width is measured) appears to be less prone to damage compared to the aculeus.

Interestingly, the phylogenetic multiple regression models indicated that the inclusion of a secondary predictor did not result in substantial enhancement of body size estimation. This is likely because all predictor traits had a strong size (length) component that was not removed prior to fitting the models. Including a secondary predictor would be more beneficial for body size prediction if that trait provided information on shape rather than just length, as body size is a composite metric incorporating length, width, and height (*Sohlström et al., 2018*; *Foerster et al., 2024a*). It is important to note that removing the size component from the predictor traits would not be appropriate in this study, as the methods provided here are designed to estimate body size based on linear measurements of body parts, under the assumption that body size is not known beforehand. Conveniently, comparing the average accuracy of simple and multiple phylogenetic regression models showed that a single predictor is typically sufficient for accurate estimations of toL and carL. This opens the possibility of performing size predictions in scorpions with minimal effort, an assumption that, to some extent, can be generalized to scorpions in general as the best-fit simple regression models resulted in accurate size estimations in non-buthid species. This is particularly suited for working with severely damaged specimens or when the identification at the family level is not possible. Yet, this finding supports the idea that the trait predictors of body size within the Buthidae clade reflect those found in the order Scorpiones. Ultimately, the flexibility of having multiple traits capable of accurately predicting body size through the simple linear equations presented in this study offers researchers a range of options for choosing the most suitable predictor traits.

## CONCLUSIONS

The predictive equations introduced in this study offer several alternatives for researchers across various fields to precisely estimate body size in scorpions, whether it pertains to total length or carapace length—the primary indicators of body size in these animals. By utilizing the methods outlined here, researchers can generate accurate estimations of body size even in scenarios where the specimens are extensively damaged or only parts of the specimens are available. Furthermore, while the focus of this study has been directed to

buthid scorpions, it has been demonstrated that the proposed predictive equations could be applicable to make educated body size predictions in non-buthid scorpions, particularly those sharing similar morphological characteristics in the examined traits and within the same size range as buthid scorpions. These may also include fossil records, though it is important to consider that size predictions for older and phylogenetically distant fossils may not be as accurate as those made for extant buthid species—an issue that is not exclusive to the present research.

## ACKNOWLEDGEMENTS

I am grateful to Luis F. de Armas, and František Kovařík for kindly and actively sending information on scorpion morphology. I extend my gratitude to all scorpion taxonomists who carefully documented the meristic data used in this study over several years of dedicated research, in special to Rolando Teruel, Wilson R. Lourenço, Luis F. de Armas, František Kovařík, Graeme Lowe, Victor Fet, and Andrés A. Ojanguren-Affilastro. I thank Toomas Tammaru (University of Tartu) and the anonymous reviewers for their valuable comments and suggestions that helped improving the manuscript.

### Funding

This work was supported by the Estonian Research Council (PRG741). The funders had no role in study design, data collection and analysis, decision to publish, or preparation of the manuscript.

### Grant Disclosures

The following grant information was disclosed by the authors:
Estonian Research Council: PRG741.

### Competing Interests

The author declares that he has no competing interests.

### Author Contributions

- Stênio Í. A. Foerster conceived and designed the experiments, performed the experiments, analyzed the data, prepared figures and/or tables, authored or reviewed drafts of the article, and approved the final draft.

### DNA Deposition

The following information was supplied regarding the deposition of DNA sequences:
The new DNA sequences are available at GenBank: PQ341278–PQ341284.

### Data Availability

The data and code are available in the Supplemental File.

## Supplemental Information

Supplemental information for this article can be found online at http://dx.doi.org/10.7717/peerj.18621#supplemental-information.

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
