# Peer review of "Body size prediction in scorpions: a phylogenetic comparative examination of linear measurements of individual body parts"

_PeerJ, doi:10.7717/peerj.18621_

## Round 0.1 · original submission · Major Revisions

Dear author,
I have now received back the comments from four reviewers. You'll see that all of them saw your work with great enthusiasm and think this could be a good addition to the literature on allometry in scorpions. Reviewers cast doubt on our hability to extrapolate your results, given you only have data for one family. This should be better explained throughout the manuscript. As for myself, I have serious doubts that the phylogenetic comparative analysis have been done correctly.

I have a couple more comments:

L. 40 classic citation would be Peters 1986 book
L. 42: dispersion -> dispersal
L. 59-66: delete
L. 138-77: Clarify if this phylogenetic hypothesis is new/different from previous ones, or if it just repeats what's already known in terms of relationship between taxa.
L. 185: I'm not sure what you mean by "a phylogenetic principal
185 component analysis (PCA) was carried out using Pagel's lambda method". There's no way to conduct a PCA with Pagel's lambda. Which version of pPCA have you used? Please, refer to Collyer and Adams 2020 Met Ecol Evol.
L. 186-8: this is not true at all. Please see Revel 2009 Evolution
L. 188-92: this is also not correct. If you're dealing with allometry using linear measurements, you need to remove the effect of size. There're at least three ways of doing it (see Klingenberg 2016). For example, you could use log-shape ratios.
L. 197-9 : delete, repetitive
L. 200: how did you calculate RMSE for an autocorrelated model?
L. 204: those are not 'multivariate' models. Multivariate refers to the nature of the response variable
L. 228-32: this happened because you didn't "remove' the effect of size on each variable, before doing the PCA
L. 299-304: delete
Do not cite figures nor tables in the Discussion.
L. 317-8: you didn't test for that
L. 336-8: this is because you didn't properly test for them. See Klingenberg 2016 and Adams and Nistri 2010
Table 1: it's really confusing having Pagel's lambda in this table. Like I said before, if you want to calculate phylogenetic signal of multivariate traits, you need Blomberg's Kmult (Adams 2014).
Table 2: how did you calcualte R2 of PGLS? This is not trivial and you should use proper methods, such as Ives 2019 Syst Biol and associated R package rr2. Also, provide dAIC, not raw AIC
Table 3: again, these are not 'multivariate' models, but akin of a multiple regression
Table 1, Fig. 3 and 4 are redundant. Keep only the PCA in Fig. 4. Projecting the phylogeny upon the ordination diagram (=phylomorphospace) would aid in the interpretation.


Reviewer 1 ·

Basic reporting

Very well written manuscript, ill all aspects.

Experimental design

Sound and clear

Validity of the findings

a phylogenetic approach that helps to understand estimations of body size in buthid scorpions

Additional comments

21: measurements instead of measures
22: correct boy for body
25: you can removed the brackets for buthid
133-135: I am not sure that I understand this sentence. You mean that some species only had two individuals?

253:254 what is the overall importance of the AIC as compared to te the RMSE values for choosing a model?

Reviewer 2 ·

Basic reporting

The main goal of this contribution is very interesting. It is well-written and requires no extended editing (except for some minor changes listed below).
The introduction provides a good overview of the relevant literature and sets the stage for the study. However, including some general background information about scorpions between lines 79 and 83 would be beneficial for a broader audience. Additionally, the rationale for focusing solely on buthid scorpions should be explained within the introduction.
Some specific sentences could be rephrased for improved clarity. Some examples where the language could be improved include lines 154-155 (e.g., replace the sentence “… plus the bothriurid species Brachistosternus paposo Ojanguren Affilastro & Pizzaro-Araya, 2014, entered as the outgroup” for with the selection of the bothriurid species Brachistosternus paposo Ojanguren Affilastro & Pizarro-Araya, 2014 as the sole outgroup), line 181 (e.g., Analyses were performed separately per sex), lines 194-195 (unclear), and lines 215-216 (clarification). Please consider these suggestions and revise the language for more clarity.
The figures and tables are informative and visually appealing. Adding details about the methods used in generating the phylogenetic tree (e.g., log-penalized method) and the phylogenetic PCA in the captions for Figures 2 and 4 would be helpful.

Experimental design

It's unclear how the authors obtained measurements from the literature (e.g., were they directly measured from images or readily available in the references?). Providing clarification on this point would be beneficial.
A major concern raised here is the discrepancy between the number of species included in the analysis (280) and the number with available measurements (158). The authors should either restrict the phylogeny to the 158 species with complete data or find a way to include data for the remaining species. For example, Centruroides serrano, with available measurements from the original description (Santibáñez-López & Ponce-Saavedra, 2009) highlights this issue.
As mentioned before, it is unclear the reasons why this study is based only on buthid scorpions. There is information available (DNA and morphometric data) for non-buthid scorpions such as bothriurids (e.g., Ojanguren-Affilastro et al., 2015), diplocentrids (e.g., Santibáñez-López et al., 2013; 2014; 2017), pseudochactids (i.e., Prendini et al., 2021, Loria et al., 2022), vaejovid scorpions (e.g., González-Santillán & Prendini, 2013; 2015a,b). A clear justification for focusing solely on buthid scorpions in this study is needed.
While Štundlová et al. (2022) used multiple outgroups, this manuscript only uses a single bothriurid outgroup. This selection can be misleading due to the potential sensitivity of topology and branch lengths to outgroup choice.
I strongly encourage including representatives of Chaeriliidae, Pseudochactidae, and the bothriurid selected here as outgroups for a more robust analysis, regardless of whether the focus remains on buthids or expands to all scorpions.
The use of a penalized likelihood method for tree conversion is appropriate, but details about the models (relaxed or correlated) and lambda values used are missing. Furthermore, the current use of only two calibration points within the "Tityus group" is insufficient. Additional points and outgroup representation are crucial for accurate dating. Alternative dating methods are suggested, such as a likelihood approximation of branch lengths using a multivariate normal distribution (implemented in PAML's baseml and mcmctree programs). This would necessitate additional outgroups and acquiring calibration points for these taxa from existing literature (e.g., Howard et al., 2019; Santibáñez-López et al., 2022).

Validity of the findings

This study presents a novel approach to analyzing scorpion morphology and represents a valuable contribution to the field. However, a few considerations could further strengthen the manuscript. These include:
Taxon selection: Focusing solely on buthid scorpions for the ingroup might limit the generalizability of the findings. Expanding the analysis to include representatives from other scorpion families could provide a more comprehensive study.
Outgroup selection: Utilizing a single bothriurid species as the outgroup could potentially influence the resulting phylogeny. Employing outgroups from multiple scorpion families (e.g., Chaeriliidae, Pseudochactidae) would likely lead to a more robust analysis.
Calibration points: The current use of only two calibration points within the 'Tityus group' might not be sufficient for accurate dating. Including additional calibration points and outgroup representation with established divergence times (as suggested in references like Howard et al., 2019 and Santibáñez-López et al., 2022) would strengthen the dating analysis.
Addressing these aspects would significantly enhance the manuscript's impact.

Additional comments

References:


González-Santillán, E., & Prendini, L. (2013). Redefinition and generic revision of the North American vaejovid scorpion subfamily Syntropinae Kraepelin, 1905, with descriptions of six new genera. Bulletin of the American Museum of Natural History, 2013(382), 1-71.
González-Santillán, E., & Prendini, L. (2015). Systematic revision of the North American syntropine vaejovid scorpions with a subaculear tubercle, Konetontli González-Santillán and Prendini, 2013. Bulletin of the American Museum of Natural History, 2015(397), 1-78.
González‐Santillán, E., & Prendini, L. (2015). Phylogeny of the N orth A merican vaejovid scorpion subfamily S yntropinae Kraepelin, 1905, based on morphology, mitochondrial and nuclear DNA. Cladistics, 31(4), 341-405.
Howard, R. J., Edgecombe, G. D., Legg, D. A., Pisani, D., & Lozano-Fernandez, J. (2019). Exploring the evolution and terrestrialization of scorpions (Arachnida: Scorpiones) with rocks and clocks. Organisms diversity & evolution, 19, 71-86.
Loria, S. F., Ehrenthal, V. L., Nguyen, A. D., & Prendini, L. (2022). Climate relicts: Asian scorpion family Pseudochactidae survived Miocene aridification in caves of the Annamite Mountains. Insect Systematics and Diversity, 6(6), 3.
Ojanguren-Affilastro, A. A., Mattoni, C. I., Ochoa, J. A., Ramírez, M. J., Ceccarelli, F. S., & Prendini, L. (2016). Phylogeny, species delimitation and convergence in the South American bothriurid scorpion genus Brachistosternus Pocock 1893: Integrating morphology, nuclear and mitochondrial DNA. Molecular Phylogenetics and Evolution, 94, 159-170.
Prendini, L., Ehrenthal, V. L., & Loria, S. F. (2021). Systematics of the relictual Asian scorpion family Pseudochactidae Gromov, 1998, with a review of cavernicolous, troglobitic, and troglomorphic scorpions. Bulletin of the American Museum of Natural History, 453(1), 1-149.
Santibáñez-López, C. E., & Ponce-Saavedra, J. (2009). A new species of Centruroides (Scorpiones: Buthidae) from the northern mountain range of Oaxaca, Mexico. Revista mexicana de biodiversidad, 80(2), 321-331.
Santibäñez-López, C. E., Francke, O. F., & Prendini, L. (2013). Systematics of the keyserlingii group of Diplocentrus Peters, 1861 (Scorpiones: Diplocentridae), with descriptions of three new species from Oaxaca, Mexico. American Museum Novitates, 2013(3777), 1-48
Santibáñez-López, C. E., Francke, O. F., & Prendini, L. (2014). Kolotl, n. gen.(Scorpiones: Diplocentridae), a new scorpion genus from Mexico. American Museum Novitates, 2014(3815), 1-14.
Santibáñez-López, C. E., Kriebel, R., & Sharma, P. P. (2017). eadem figura manet: Measuring morphological convergence in diplocentrid scorpions (Arachnida: Scorpiones: Diplocentridae) under a multilocus phylogenetic framework. Invertebrate Systematics, 31(3), 233-248.
Santibáñez-López, C. E., Aharon, S., Ballesteros, J. A., Gainett, G., Baker, C. M., González-Santillán, E., ... & Sharma, P. P. (2022). Phylogenomics of scorpions reveal contemporaneous diversification of scorpion mammalian predators and mammal-active sodium channel toxins. Systematic Biology, 71(6), 1281-1289.
Štundlová, J., Šťáhlavský, F., Opatova, V., Stundl, J., Kovařík, F., Dolejš, P., & Šmíd, J. (2022). Molecular data do not support the traditional morphology-based groupings in the scorpion family Buthidae (Arachnida: Scorpiones). Molecular Phylogenetics and Evolution, 173, 107511.

·

Basic reporting

Please see attachment

Experimental design

Please see attachment

Validity of the findings

Please see attachment

Reviewer 4 ·

Basic reporting

Dear Author,
I have reviewed the manuscript titled "Body Size Prediction in Scorpions: A Phylogenetic Comparative Examination of Linear Measurements of Individual Body Parts" (PeerJ #100839). The paper presents novel data on the sex-specific predictive power of six linear measurements in bi- and multivariate phylogenetic comparative settings. The comparative analysis and the inclusion of multiple potential predictors are strong aspects of this work and will be extremely useful for further testing hypotheses in scorpion models and for methodological proposals concerning the measurement of fossil material. Overall, I find the manuscript very interesting, well-written, and detailed in its methodology. However, despite the authors' considerable effort in conducting phylogenetic studies, the discussion of the phylogenetic signal found in the analyses and the factors driving the differences between sexes in the optimal models is not thoroughly explored. Nevertheless, I recommend this manuscript for publication after addressing the following issues.

Basic reporting
I would like to congratulate the authors for this work, which is excellently written in clear English, with excellent readability. The introduction and discussion are fluently articulated. The manuscript is well-supported by sufficient literature references and provides an excellent background. The structure of the manuscript is appropriate, with adequate figures and tables (although see comments below), and the raw data is available.

L42. Maybe better "dispersal capability" than "dispersion capability".
L49-50. Could you provide more examples beyond Foerster et al (2024) where body weight is currently used as a proxy for body size?
L51-54. Beyond the “practical” or “methodological” disadvantages of using weight as a size indicator, what other disadvantages (at a theoretical level) can you mention?
L77. For consistency, remove comma after author in citations: Huxley, 1932; Packard, 2013.
L90-91. Reorder citations chronologically.

Experimental design

The research question is clearly defined and relevant; however, the manuscript does not state any specific hypotheses or predictions. Did the authors have any predictions about which predictor might be the most effective, based on prior knowledge of the species or previous studies? Additionally, did they anticipate that the predictors might differ between sexes for any reason? Were there expectations regarding the influence of phylogeny on the estimates? The research has been conducted rigorously and to a high technical standard. The methods are generally thorough, though several details need clarification for a complete understanding.

Experimental design
L126. Be more specific regarding “the orientations of Sissom et al. (1990).”
L128-131. I understand the justifications for analyzing the sexes separately. However, I wonder if it would have been possible to include all the data in the PGLS models together and add sex as other predictor (in interaction with character predictor), analyzing the differential effect of sex on the estimates. I feel that this approach might have been more rigorous than analyzing the datasets separately. Why was this not done?
L133-135. “Species means were then calculated for an average of two individuals per species in each data set.” What exactly is meant by “two individuals per species”? In Supplementary Information 1, I see species with only one individual, as well as cases with many more individuals. Perhaps this point could be clarified better. I understand that averages were used in subsequent models, but is it possible to include sample size or standard deviation in some way? For species where only one individual was measured, the precision of the estimate would not be the same as for those where 28 individuals were considered. How was the possibility of unaccounted intra-specific variability addressed?
L137. If I’m not mistaken, in the context of this work and its objectives, the phylogeny of buthids was created solely to perform a phylogenetic correction in the models predicting total body size and prosoma size. Looking at Supplementary Material 1, I see that there are 121 species without associated morphological measurements (they are only present in the phylogeny). What is the purpose of including these species in the phylogenetic analysis? What information do they contribute to this work?
L182-183. I was unable to find any mention of the nature of the distribution of the variables considered, only that they were logarithmically transformed. Were they all normal initially? I see that the Pearson correlation coefficient was used, but if normality was not met, the Spearman coefficient should be used.
L183-192. I must confess that I got a bit lost here with the PCA. I understand that PC1 captured most of the variability and was loaded by size variables. Still, the interpretation of “shape” made by the author is confusing to me. For example, in a geometric morphometrics study, one might assert that PC2 explains shape (non-orthogonal to size in PC1). But here, where only some size variables were considered, why is PC2 said to represent shape? Could it capture variability of other unconsidered variables that are not orthogonal to the analyzed ones? I would appreciate an explanation on this point.
L194-195. A query about terminology: generally, the term “multivariate” is used when there are multiple response variables in the same model. Here, there is only one response variable: total body length or carapace length (in independent models) and two predictors. It doesn’t seem correct to call these “multivariate PGLS." They are PGLS with one or two predictors; please review the terminology between bivariate and multivariate, as it seems somewhat confusing. If terminology is modified, do so consistently throughout the manuscript. If the author maintain their terminology, please cite a work that uses this terminology.
L199-201. It is reported that model performance is evaluated by RMSE, but it would be helpful to add that “lower RMSE values indicate better model fit.” Additionally, I see that Tables 2-7 report the coefficient of determination (R²) and AIC. The methodology should mention that these parameters were calculated or the joint use of these parameters to evaluate the best predictive model. Similarly, the tables report Lambda, but it would be useful to clarify in the methodology that values close to 1 suggest a strong phylogenetic signal, implying that phylogeny is important for explaining residual data.
L201-206. In the correlation analyses, strong positive relationships were found between all parameters. You are including two predictor variables that might be collinear. Multicollinearity inflates the variances of coefficient estimates, making them less precise. When two predictors are highly correlated, it becomes challenging to determine the individual effect of each predictor on the response variable. How did you address this issue when running the models?
L215-216. Check whether these packages have the same citation or if they need to be cited separately.

Validity of the findings

All data have been provided; they are robust, statistically sound, and controlled. The results accurately present the findings from the data analyses performed. The discussion is well outlined, although I believe that some topics could be discussed more effectively (see below). I emphasize that the work focuses on the comparative approach (which is its strength), but nothing is mentioned about this point in the results or discussion. The importance of the work in the field of scorpions and its application to other models could be highlighted a bit more. Conclusions are well stated, linked to the original research question, and limited to supporting results.

L223-226. Although I assume all correlations were statistically significant, is there a reason why the p-values associated with each R value were not reported? In Fig. 3, the author could include the p-values in the upper diagonal.
L230-233. See my previous comment about the justification for discussing "shape" here. What shape variables do you interpret as being represented by PC2? You mention that smaller and "stout" species are related to positive values of PC1; does "stout" refer to shape or which variable are you associating it with? What does it mean in terms of the measured parameters?
L305-311. This work did not evaluate the predictive power of carapace length for total body size in a comparative framework. The correlations performed did not include phylogenetic corrections, so the assertion that “it constitutes the first empirical validation of the suitability of carapace length as a proxy of body size in these animals, tested with robust comparative data” may not be correct. What can be said is that “certain body structures were found to be good predictors of body size or carapace size in a comparative framework.” However, it would have been very useful to include a model that assessed the predictive power of carapace length for total body length to address: a) how good it is as a proxy for total size in buthids (in a comparative framework), and b) how different from the most commonly used proxy is the use of other characters to estimate body size. In this way it would be possible to estimate the performance of other indicators of body size in relation to the performance of carapace length.
L311-316. “With a few exceptions (McLean et al. 2018) the use of carapace length as an indicator of body size is not always clear.” I suggest rephrasing to “With a few exceptions (McLean et al. 2018), the use of carapace length as an indicator of body size is not always properly justified.” However, I repeat that your work does not do this either; you only show a correlation between carapace length and total length without phylogenetic correction (which even shows that for males, the R² is higher for other structures than for carapace length).
L336-342. I believe the topic of allometry could be explored much more in your work. The slope between body length/carapace length and the length of certain characters indicates the allometric pattern. By incorporating a predictor variable “sex” into the models, you could determine if the slopes differ between sexes and assess allometric patterns. I think this discussion is very important, especially considering that you analyzed sex-specific models. Additionally, you found differences between sexes in the best predictors of carapace size. What could explain these differences? What selective pressures might explain this type of dimorphism?
L343. Although I recognize (and admire) the author’s substantial work in constructing the phylogeny of these species, I believe it is not fully utilized in the context of this work. The focus is comparative, but nothing is mentioned about the phylogenetic signal found in the models, nor is it interpreted what it might inform about the the evolutionary trajectories of the relationships between the size of different traits in these species. I understand that the work is a methodological contribution, but since you did such extensive work to include a phylogenetic correction, it should be mentioned in the discussion. Additionally, the phylogeny figure as it is could easily be placed in the supplementary material, as it does not contribute to the main idea of the work. You could have mapped body size or mapped the residuals (errors) along the phylogenetic tree to identify patterns or biases in the predictions.
L401. I think the author can elaborate a little more about the relevance and importance of the findings and their potential application to other scorpion families or other models.

Additional comments

Tables and Figures
Table 1: I suggest replacing “for by using the lambda method” with “using Pagel’s Lambda method.”
Tables 2-7, S2-24: Indicate what R² and AIC mean in your models (even though these terms are well-known, they should be clarified just as you have done with Lambda, RMSE, and L/U).
Figure 2: See my previous comment; as it stands, this figure should be moved to the supplementary material.
Figure 3: See my previous comment about adding the p-values.
Figure 4: Replace “Dim1” and “Dim2” with “PC1” and “PC2,” respectively. If possible, maintain the same scale on the X-axis for graphs “a” and “b.”

References
L477-480. If necessary, reorder chronologically: Hódar, J. (1997). and Hódar, J. A. (1996).
L536-546. If necessary, reorder chronologically: Lira et al. (2023); Lira et al. (2018); Lira et al. (2021).
L665-673. If necessary, reorder chronologically: Teruel, 2014; Teruel, R., & Rein, J. O. (2010); Teruel, R., & Stockwell, S. A. (2002).

---

## Round 0.2 · Minor Revisions

Your manuscript has now been seen by two out of the three reviewers of the previous round. While R2 is happy with this new version and only has a few suggestions on how you presented the phylogenetic tree inferred, R4 has some important concerns on the presentation of results and data analysis, of which I tend to agree almost 100%.

Also, avoid extrapollating too much your results to Scorpions as a whole.
As R4 points out, the statement in L. 219-20 is not correct. Simply including species as random effect (also, what kind of random effect, random intercept?) doesn't control for phylogenetic autocorrelation. If you want to continue using MCMCglmm, which I don't think you need to, you need to include the inverse of the phylogentic distance matrix in the model. See p. 112 of the package vignette (http://cran.nexr.com/web/packages/MCMCglmm/vignettes/CourseNotes.pdf). But I really don't think you need to use MCMCglmm, because it adds more - but unnecessary - complexity to the analysis. For example, you didn't inform which priors you used, how many chains you run, for how long, the interval of their sampling, how much burn-in was used, how you diagnosed the model, how you diagnosed the MCMCs...

i also agree with R4 that you can't directly compare models with different predictors using RMSE, so you still need to bring AICc back. Or even better DIC and/or Bayes Factor, which work better in a Bayesian context.

You also can't interpret PC1 and PC2 the way you did in L. 214-5, see Revell 2009 Evolution. Worth citing Collyer & Adams 2021 Met Ecol Evol when talking about pPCA too.

Reviewer 2 ·

Basic reporting

The revised manuscript has significantly improved upon the previous version. The additional context and references have enhanced the clarity and understanding of the research. While the primary focus of the study is not on the evolution of buthid scorpions, a brief description of this topic within the results section would provide valuable insights and strengthen the overall contribution of the work. This can be accomplished as "The phylogeny and divergence times obtained here are congruent with others (i.e., Stundlova et al. 2022; Santibanez-Lopez et al., 2022).

Experimental design

The author has effectively addressed my previous concerns and provided more detailed explanations of the new methodologies employed in this version.

Validity of the findings

The study's novel approach to analyzing morphology, particularly for groups with limited museum specimens, is commendable. To further enhance the manuscript, I recommend including a brief overview of the phylogeny obtained in this study. This would provide a complete picture of the research findings and their implications (e.g., comparing the obtained topology against others).

Reviewer 4 ·

Basic reporting

Dear Author,
I have reviewed the new version of the manuscript “Body size prediction in scorpions: a phylogenetic comparative examination of linear measurements of individual body parts.” This paper presents novel data on the predictive power of seven linear measurements within a phylogenetic comparative framework. I believe this is a significant methodological contribution to the study of scorpion morphology. I am pleased with the revisions the author has made, which have greatly improved the manuscript, particularly in sections of the methodology that were previously lacking in detail. The empirical demonstration of size predictions in non-buthid species was a particularly valuable addition, as it broadens the generality of the study's results. Furthermore, the author’s effort to make the implementation of these models more accessible through R functions is commendable.
However, I still have some concerns regarding the chosen methodology for generating the predictive equations (see below). Otherwise, I have a few minor suggestions to clarify certain aspects of the paper, but overall, I believe that after these revisions, the paper will be ready for acceptance.

Experimental design

1.First, I should clarify that I’m not familiar with these models or the underlying statistics. Apologies if I have misunderstood something, but I believe that a point needs to be clarified. The author states on L218, “This was achieved using Bayesian mixed linear models implemented in the MCMCglmm R package”, they later refer to “Bayesian phylogenetic mixed linear models.” While I understand the intention was to incorporate the interaction term of sex*trait and species as a random effect, what is the rationale for using this new Bayesian approach? Is it not possible to perform PGLMMs directly (e.g., using the phyr package)? If the author is not incorporating phylogenetic signal into the predictive models (in addition to the species-level random effect), which are the primary contribution of the work, it would not constitute a “phylogenetic comparative examination” as indicated in the title…

2. In the previous version, the manuscript presented RMSE and AIC values, and based on this combined analysis, it was possible to conclude which predictive model was “better.” AIC has an accepted criterion in the scientific community, where a ΔAIC of 2 is considered an acceptable difference. In this new version, only RMSE was retained, and it seems AIC is no longer supported by the new models. However, what criterion is being used to compare two RMSE values and conclude that one model is better than another? I understand that a lower RMSE indicates a better model, but what would be the “tolerance” between two RMSE values that would indicate both models are equally good, despite differing by decimals? For instance, the author concludes that Met5L is a better predictor of body size than CarL due to a 0.05 RMSE difference. Is it sufficient to say one trait is a better predictor than another? Then the author compares simple and multiple models for predicting size and finds a difference in RMSE of 1.28, which is now considered minimal and negligible, favoring simple models. Again, what is the standard for deciding that the first difference is sufficient and the second is not?

3. Shape in the context of this manuscript remains somewhat confusing to me. The aim of the study is to determine whether the size (length) of a predictor trait can reliably predict body size (length). The objective does not seem to involve shape in any way, and shape is neither measured nor included in any of the analyzed variables. On L405, the author states, “body size is a composite metric incorporating length, width, and height,” but in this work the response variable has consistently been “length”.
Therefore, I don't understand how measuring trait lengths to predict body length can yield any conclusions about shape, or how predictor traits could load onto a PC that represents shape. To study shape, a different approach (e.g., geometric morphometrics) or different types of variables (e.g., morphometric ratios) would be required. While I understand the author’s explanation regarding non-orthogonality, I was not surprised that there was no result for shape because it was never measured. I leave it to the author to decide whether to include further clarification on this in the manuscript.
Lastly, on L249-251, the author continues to refer to “smaller and stout species ... while larger and slender species ...” Stout and slender seem to describe shape traits; otherwise, why distinguish between smaller and larger species if these are synonymous? I find this confusing. For example: in Konuma et al. 2011 PC2 represents the average ‘shape’ as the combination of length/width and refers to stout and slender... but that is not the size.

Validity of the findings

no comment

Additional comments

Minor comments:
• There should be more consistency in the use of abbreviations for morphological traits (e.g., totL, carL). Once an abbreviation is introduced, it should be consistently used throughout. There is a lot of inconsistency in their usage (sometimes they are used, sometimes not). For figure and table captions, please move the abbreviation after the trait name.
• L117-119: Avoid repeating “retrieved” in two consecutive sentences.
• L231-233: “Lastly, the best-fit simple regression models were applied for predicting total length and carapace length in 30 randomly selected non-buthid scorpions.” This anticipates the choice of simple models over multiple regression models for buthids. I suggest adding a preceding sentence such as, “Based on the results obtained with buthids, where the best predictions of body size were achieved with simple regression models (see results), we applied the best-fit simple regression model to predict total length and carapace length in 30 randomly selected non-buthid scorpions.” Otherwise, it raises the question of why multiple regression models were not also tested with non-buthid scorpion species.
• L233: Consider changing “non-buthid scorpions to access the applicability” to “assess the applicability.”
• L343-351: I agree with the author’s explanation regarding the role of sexual selection in explaining the greater morphological dispersion of males in two-dimensional trait space. However, this does not explain why the dispersion is greater in smaller species. I suggest adding a line to address this to complete the argument.
• L401-403: “This is likely because all predictor traits had a strong size (length) component that was not removed prior to fitting the models.” The logic is clear, but the explanation is somewhat confusing. Could this be rephrased for clarity?
• L415-417: “Yet, this finding supports the idea that the morphological diversity within the Buthidae clade mirrors that found in the order Scorpiones.” This seems like a bold statement. While I understand that the selected predictor for buthids also worked well for non-buthids, the morphological diversity across families varies significantly due to a range of selective pressures. I would suggest phrasing this more conservatively, such as: “Yet, this finding supports the idea that the trait predictors of body size within the Buthidae clade reflect those found in the order Scorpiones.”

Figure 2: “The gradient in size along axis 1 is depicted by the proximity of trait vectors to axis 1 and the linear variation in total length values across axis 1.” This is somewhat unclear and could benefit from clearer phrasing. Also, out of personal curiosity, is there a species where females are markedly separated along PC2? What can be inferred from this, and which species is it?

Figure 3: Consider changing axis labels from (In mm) to (mm) or removing the capital letter.

Table 2: Add “CI indicates the confidence intervals (L = lower limit, U = upper limit).” ESS (effective sample size obtained from the posterior MCMC samples of the model estimates) is mentioned for the first time in this table. This should be introduced earlier in the methods, and pMCMC should also be clarified in the methods and referenced in the table caption. In place of “response,” consider using “response variable,” and instead of “Parameter,” “predictor trait” might be more appropriate. Where the trait name is listed, it might be better to specify “slope,” so it reads “intercept” and “slope,” indicating which trait is being referred to.

Table 3: Consider changing “Size predictions for non-buthid scorpions (n = 30)” to “Size predictions for non-buthid scorpions (n = 30 species)” and organize the species by some criteria—alphabetically or by family. The term “deviance” is mentioned for the first time in this table. It should be introduced earlier in the methods. I find it difficult to contextualize what constitutes a large or small deviance. I see low values (e.g., 0.09) and high values (e.g., 7.47), but this does not appear to be discussed or compared with deviance values for buthids. I feel this comparison with non-buthids is important, but it doesn’t seem fully addressed.

Supplementary Information: Interaction models: Why does this table report estimates for the interaction between trait and sex for males? I understand it shows the differential p-value with the interaction for females, but it might be clearer to report the direct p-value for the trait interaction for both sexes (ln_trait*sex).

---

## Round 0.3 · accepted · Accept

Thank you for addressing all comments and engaging with reviewer's suggestions. I'm happy to recommend the manuscript for publication as is